# A Review of the Potential of Poly-(lactide-co-glycolide) Nanoparticles as a Delivery System for an Active Antimycobacterial Compound, 7-Methyljuglone

**DOI:** 10.3390/pharmaceutics16020216

**Published:** 2024-02-01

**Authors:** Bianca Diedericks, Anna-Mari Kok, Vusani Mandiwana, Namrita Lall

**Affiliations:** 1Department of Plant and Soil Sciences, University of Pretoria, Pretoria 0002, South Africa; u17023409@tuks.co.za (B.D.); annamarikok@gmail.com (A.-M.K.); 2Research Fellow, South African International Maritime Institute (SAIMI), Nelson Mandela University, Gqeberha 6019, South Africa; 3Chemicals Cluster, Centre for Nanostructures and Advanced Materials, Council for Scientific and Industrial Research, Pretoria 0001, South Africa; vmandiwana@csir.co.za; 4School of Natural Resources, University of Missouri, Columbia, MO 65211, USA; 5College of Pharmacy, JSS Academy of Higher Education and Research, Mysuru 643001, India; 6Senior Research Fellow, Bio-Tech R&D Institute, University of the West Indies, Kingston IAU-016615, Jamaica

**Keywords:** tuberculosis (TB), antimycobacterial, cytotoxicity, 7-methyljuglone, poly-(lactide-co-glycolide) (PLGA), nanoparticle

## Abstract

7-Methyljuglone (7-MJ) is a pure compound isolated from the roots of *Euclea natalensis* A. DC., a shrub indigenous to South Africa. It exhibits significant promise as a potential treatment for the highly communicable disease tuberculosis (TB), owing to its effective antimycobacterial activity against *Mycobacterium tuberculosis*. Despite its potential therapeutic benefits, 7-MJ has demonstrated in vitro cytotoxicity against various cancerous and non-cancerous cell lines, raising concerns about its safety for consumption by TB patients. Therefore, this review focuses on exploring the potential of poly-(lactide-co-glycolic) acid (PLGA) nanoparticles as a delivery system, which has been shown to decrease in vitro cytotoxicity, and 7-MJ as an effective antimycobacterial compound.

## 1. Introduction

Tuberculosis (TB) is a highly infectious disease caused by the bacteria *Mycobacterium tuberculosis*. Tuberculosis is a complex and communicable disease that was the global primary source of death caused by a solitary infectious agent, which ranked above human immunodeficiency virus (HIV)/acquired immunodeficiency syndrome (AIDS) until 2019, when the coronavirus (COVID-19) pandemic broke out [1,2]. The COVID-19 pandemic has since caused a significant setback to the many years of headway in providing vital TB services and lowering the disease burden [2].

Using natural products, such as plants, as alternative therapies may assist in improving the therapeutic efficacy and, in some cases, can decrease some of the side effects experienced due to conventional drugs [3,4]. Medicinal plants are often promoted as natural and therefore harmless, which is true in most cases; however, they are not always free from adverse effects or toxicity [5]. Many very active medicinal plants and pure compounds isolated from medicinal plants have been found to possess a degree of toxicity [6]. This review elaborated on the efficacy as well as toxicity of an active antimycobacterial pure compound, 7-methyljuglone (7-MJ).

Loading natural products into nanoparticles as drug delivery systems for the treatment of diseases has been widely explored with various aims, such as to guard sensitive compounds against degradation, increase their bioavailability, and decrease the cytotoxicity of toxic compounds [7]. Literature indicates that using poly-(lactide-co-glycolide) (PLGA) nanoparticles as a delivery system has shown the potential to reduce the toxicity of known toxic compounds. Therefore, this review article evaluated the literature on PLGA nanoparticles used as drug delivery systems for toxic hydrophobic drugs similar to 7-MJ. The aim of this evaluation was to determine if PLGA nanoparticles would be a suitable drug delivery system for the toxic hydrophobic drug 7-MJ.

### 1.1. Impact of COVID-19 on TB

In 2019, a highly infectious viral epidemic called coronavirus disease-2019 (COVID-19) broke out in China, resulting from infection with severe acute respiratory syndrome coronavirus 2 (SARS-CoV-2) [2]. In South Africa, the COVID-19 pandemic first emerged during the early parts of 2020, leading to nationwide lockdowns being implemented. These lockdowns had a negative impact on TB patients and TB-related services. With the subsequent increase in COVID-19 infections, actions to address the emergent disease became a priority across all government divisions, resulting in health workers and supplies being redirected away from routine services. The decreased supplies and attention to support TB patients led to fewer TB diagnoses [8]. This, in turn, can result in subordinate treatment outcomes and, subsequently, an increase in transmission, causing a rise in the current TB burden and TB-related deaths. It was also found that patients currently infected with TB or those with a history of TB infection were more vulnerable to COVID-19 infection [9].

The most significant impact that the COVID-19 pandemic had on the global TB burden was the substantial decline in the number of reported patients newly diagnosed with TB infection. This decline was likely a result of disruptions to TB health service delivery and changes in the demand for TB diagnostic and treatment services, attributable to quarantine, isolation, and travel restrictions implemented in an attempt to mitigate the pandemic [10].

An example of the pronounced drop in diagnosed and reported patients is evident in the sharp 18% decline in newly reported TB cases from 2019 to 2020. In 2019, 7.1 million people were reported to be diagnosed with TB, whereas in 2020, only 5.8 million people were subsequently reported to be diagnosed. Instances of TB deaths increased due to a decline in access to TB diagnosis and treatment. In 2020, the best estimates of TB deaths were 1.3 million, with a further 214,000 deaths among patients co-infected with HIV. Other impacts included the decrease in global expenditure on TB diagnostics, reductions in the number of patients receiving precautionary TB treatment (from 3.6 million to 2.8 million), and reductions in the number of patients receiving treatment for drug-resistant TB (from 177,100 to 150,359) between 2019 and 2020 [2].

### 1.2. Current Challenges in TB Treatment

The standard first-line drugs, as shown in Table 1, are taken for 6–9 months and are not effective for patients infected with drug-resistant *M. tuberculosis* strains. For these patients, more prolonged treatment periods of up to 18–24 months are required, which can lead to poor adherence, higher costs, and increased toxicity [11].

The long duration of TB treatment and the associated expenses are among the primary reasons many TB patients either cannot or choose not to seek diagnosis or treatment. Instead, patients often explore alternative methods such as acupuncture, homeopathy, herbs, and supplements to address their symptoms. This underscores the importance of identifying more cost-effective and accessible treatment regimens. The persistent increase in drug-resistant TB and the decline in global spending on TB resources further underscore the necessity for developing successful alternative or adjunctive therapeutic methods, such as immunomodulatory therapies used in conjunction with antimicrobial treatments. These approaches can help reduce the duration and enhance the efficacy of the current TB treatment regimens [15,16]. Implementing novel therapeutic methods can effectively combat the extensive levels of antibiotic resistance prevalent in the global healthcare sector. Additionally, it has the potential to alleviate some of the financial strain on global spending for TB treatment, aiming for a treatment period ideally shorter than the current 6–9 months [17].

### 1.3. Plants as Alternative Care

Using natural products, such as plants, as alternative or adjunctive therapies may assist in improving therapeutic outcomes and decreasing some of the side effects experienced due to conventional drugs [3]. Medicinal plants have been utilized for decades and play a major role in traditional medicine. Traditional medicine is defined as the totality of knowledge, skills, and procedures grounded in the beliefs, experiences, and ideas indigenous to various cultures. It is utilized to sustain wellbeing and to identify, prevent, treat, or improve physical and mental diseases [18]. The significance of pharmacological natural plants lies in their potential to serve as starting materials for synthesizing drugs or to be directly used as therapeutics. They can also act as representatives of pharmacologically active compounds that may be less toxic or exhibit higher activity than their synthetic equivalents [19]. Traditional medicine is one of the most vital healthcare systems in Africa. Traditional African medicine has evolved based on healing systems and serves as a key component in drug discovery and pharmacology. Therefore, within the drug discovery process, traditional African medicine can contribute to many breakthroughs [20,21].

#### 1.3.1. Introduction to *Euclea natalensis* A. DC.

One such medicinal plant indigenous to South Africa, *Euclea natalensis* A. DC. (*E. natalensis*), has shown great cultural significance and traditional use as a potential therapeutic intervention in TB management [22]. Indigenous people throughout Southern Africa, particularly in the East and South Coast regions extending outward up to Mozambique, Swaziland, and Ethiopia, have been using this deciduous tree for the treatment of respiratory and dermatological ailments [22,23]. *Euclea natalensis* is traditionally employed as a medicine in 57% of the countries where it is indigenous [24]. In a study conducted by Lall et al. (2016), the ethanolic shoot extract of *E. natalensis* exhibited a minimum inhibitory concentration (MIC) value of 125 µg/mL, indicating moderate antimycobacterial ability [22]. This was further supported by in vivo studies where the mycobacterial load in infected mice decreased when treated with the ethanolic shoot extract [22]. *Euclea natalensis* belongs to the Ebenaceae family and is commonly known as Natal Ebony. This medicinally active plant is neither threatened nor endangered, and it is widely distributed. Therefore, the sustainability of the plant will not be immediately threatened by research initiatives conducted in a responsible manner [23].

#### 1.3.2. Pure Compounds Isolated from *Euclea natalensis* A. DC.

Throughout the Ebenaceae family, in species such as *Euclea natalensis* A. DC., *Diospyros mespiliformis* Hochst., *Diospyros ferrea* (Willd.) Bak., and *Diospyros tricolor* (Schumach. and Thonn.) Hiern, the presence of naphthoquinones is widespread, and these compounds possess significant antitubercular activity, among other properties. Five of the most biologically active naphthoquinones isolated from the root of *E. natalensis* include diospyrin, neodiospyrin, isodiospyrin, shinanolone, and 7-methyljuglone (7-MJ) [22]. Among all the pure compounds isolated from *E. natalensis*, 7-MJ exhibited the most promising antimycobacterial activity [22].

#### 1.3.3. 7-Methyljuglone as a Potential Antimycobacterial Therapeutic Agent

The 7-MJ isolated from the root chloroform extracts of *E. natalensis* exhibited MIC values of 0.50 µg/mL against *M. tuberculosis* and 1.57 µg/mL against *M. smegmatis* [25,26]. The crude *E. natalensis* chloroform root extracts had MIC values of 8 µg/mL and 7.33 µg/mL on *M. tuberculosis* and *M. smegmatis*, respectively [25]. Intracellularly, 7-MJ showed an EC_90_ (90% maximal effective concentration) of 0.57 µg/mL for the growth inhibition of *M. tuberculosis* in mouse macrophage (J774A.1) cells [27]. The inhibitory effect of 7-MJ on *M. tuberculosis* intracellularly (EC_90_) in J774A.1 cells and extracellularly (MIC) is comparable to that of streptomycin (EC_90_ = 1.11 µg/mL and MIC = 0.625 µg/mL) and ethambutol (EC_90_ = 1.62 µg/mL and MIC = 1.25 µg/mL). In a synergistic study, it was shown that 7-MJ also has the ability to improve the activity of isoniazid and rifampicin, as they showed fractional inhibitory concentrations of 0.25 and 0.5, respectively, with 7-MJ [27,28]. 7-Methyljuglone (Figure 1A) is a monomer of diospyrin (Figure 1B), wherein diospyrin showed a MIC value of 100 µg/mL against *M. tuberculosis*, which is 200-fold higher when compared to 7-MJ [29]. Due to the effective antimycobacterial activity that has previously been found, the current review will mainly focus on naphthoquinone (i.e., 7-MJ) as a potential antimycobacterial for TB treatment.

#### 1.3.4. Mechanism of Action of 7-Methyljuglone

7-Methyljuglone has a chemical structure very similar to that of menaquinone. Menaquinone is a natural redox cycler found in the Mycobacterium family. It is responsible within the respiratory chain for mediating electron transfer between different membrane-bound enzymes [30,31]. Mammals and most bacteria make use of ubiquinone to fulfill the function of electron transport. *Mycobacterium tuberculosis*, however, lacks ubiquinone and only has the ability to utilize menaquinone in its electron transport chain. This makes it an appealing drug target, seeing as it lacks a human homologue [32]. In a study carried out by Van der Kooy et al. (2006), it was postulated that the mechanism of action of 7-MJ is that of an inhibitory interaction with the enzymes found within the mycobacterial electron transport chain. Due to the structural similarities found between 7-MJ and menaquinone (Figure 2), the electron flow can then potentially be reduced or halted due to the imbalance in the redox potential through the incorporation of 7-MJ [26].

Another possibility is that 7-MJ can bind to the Men enzymes (MenA [1,4-Dihydroxy-2-naphthoate isoprenyltransferase], MenB [1,4-Dihydroxy-2-naphthoyl-CoA synthase], MenC [O-Succinylbenzoate synthase], MenD [2-Succinyl-5-enolpyruvyl-6-hydroxy-3-cyclohexadiene-1-carboxylate synthase], MenE [O-Succinylbenzoate synthase], MenF [Isochorismate synthase], MenG [Demethylmenaquinone methyltransferase], MenH [Demethylmenaquinone methyltransferase], and MenI [1,4-Dihydroxy-2-naphthoyl-CoA hydrolase]), which are responsible for the synthesis of menaquinone and therefore inhibit the addition of the hydrophobic sidechain [26,33]. This inhibition will influence the production of adenosine triphosphate (ATP) and lead to a detrimental effect on the bacterium [26].

#### 1.3.5. The Sustainability of 7-Methyljuglone

There is currently concern regarding the sustainable availability of 7-MJ, as a study performed by Lall et al. (2005) reported a very low yield of 0.03% isolated from the root of *E. natalensis* [27]. In literature, leaves are generally removed from the aerial parts since the leaves are not used by the indigenous communities for the treatment of TB. There is currently no literature that suggests that other plant parts of *E. natalensis* have any significant antimycobacterial activity [24,34]. This can potentially place pressure on the current *Euclea* populations due to the roots being the most bioactive plant part identified and the plant part where 7-MJ is predominantly found. This has since led to the artificial synthesis of 7-MJ and its derivatives, of which the activity on *M. tuberculosis* was compared to that of the parent compound in a structure-activity bioassay. However, out of the 19 derivatives tested, 7-MJ was still the most active and selective antitubercular agent [31]. This indicates that 7-MJ can be considered as a potential antimycobacterial drug. However, in a study carried out by Kishore et al. (2014), 7-MJ has been shown to be cytotoxic to two human cell lines, namely, peripheral blood mononuclear cells (PBMCs) and human macrophages (U937) [35].

#### 1.3.6. The Cytotoxic Effects of 7-Methyljuglone

7-Methyljuglone has been shown to have very promising antimycobacterial properties; however, 7-MJ has also been shown to be cytotoxic to various cancer and non-cancer cell lines, as summarized in Table 2. A compound that exhibits a half-maximal inhibitory concentration (IC_50_) of less than 10 μM is reasoned to have in vitro cytotoxic activity against cancer cells [36].

In a study conducted by Kishore et al. (2014), the exact cytotoxic IC_50_ value of 7-MJ on U937 cells was not reported; however, it was reported that the IC_50_ lies between 1 and 5 µg/mL (5.31 and 26.6 µM) [35]. 7-Methyljuglone can be regarded as a promising cytotoxic compound against respective cancerous cell lines such as DU145, KB, Lu1, LNCaP, and HL60 [35,37]. It, however, does not seem to have selectivity toward cancerous cell lines in comparison with non-cancerous cell lines, as it showed similar in vitro cytotoxicity against most of the non-cancerous cell lines previously tested. The lack of selectivity exhibited by 7-MJ implies that this drug will not be effective in targeting cancer cells specifically and is therefore not a good contender for cancer treatment. Achieving selectivity for cancer cells is a key goal in the drug development of anti-cancer drugs: to maximize efficacy while minimizing toxicity and adverse effects on normal tissues [38]. For the use of 7-MJ as a potential anti-TB drug, the selectivity of cancerous versus non-cancerous cells does not influence its efficacy but does raise drug safety concerns [39]. Its cytotoxicity toward the HUVEC cell line also suggested that pregnant women would need to practice caution when taking 7-MJ [37].

With the focus on reducing the cytotoxic effect of 7-MJ, nanotechnology may be a promising alternative to consider. As a system to deliver therapeutic agents, nanotechnology can provide advantages such as drastically reducing the size of the drug taken up, which will result in a higher surface-to-volume ratio, protecting the drug moiety within the nanoparticle against degradation, and reducing the toxic effects of the therapeutic agent [40].

### 1.4. Nanoparticles and Their Role in Drug Development

With the rise of innovative nanotechnologies, there are currently a myriad of different ways to prepare nanoparticle formulations, specifically anti-TB drug delivery systems. Some examples of different nanocarriers previously explored for anti-TB therapy are shown in Figure 3, which includes liposomes, nanoparticles, polymetric micelles, polymersomes, and niosomes [41]. In a study by Donnellan et al. (2017), solid-drug nanoparticles were synthesized as a drug delivery system for rifampicin. In the in vitro study, this nanoparticle formulation showed a 50-fold increase in antimycobacterial efficacy compared to that of free rifampicin [42]. In another study conducted by Hanieh et al. (2022), nanoemulsions and niosomes of novel mycobacterial membrane protein large inhibitors, BM625 and BM819, exhibited promising antimycobacterial activity against M. tuberculosis [43].

Nanoparticles are <1000 nm in diameter and are colloidal particles that are submicron-sized [44,45,46]. The physiochemical properties of nanoparticles, such as their composition, hydrophobicity, charge on their surface, and size, impact their eventual immunogenicity, cellular uptake, biodistribution, drug-loading capacity, and cellular uptake [47].

A compound can either be attached, encapsulated, entrapped, or dissolved in a nanoparticle matrix [48]. According to the composition of nanoparticles, they can be divided into different groups, for example, lipid-based, polymeric-, semiconductor-, metal-, carbon-based, and ceramic nanoparticles [49].

A particularly interesting polymeric nanoparticle-based delivery system for oral drug delivery due to its safe biodegradation products, commercial availability, degradability in physiological environments, and biocompatibility is poly-(lactide-co-glycolic) acid (PLGA) [50,51]. Poly-(lactide-co-glycolic) acid is made from two monomers: poly lactic acid (PLA) and poly glycolic acid (PGA), which can also influence the properties of the PLGA [52]. Poly-(lactide-co-glycolic) acid copolymers have been indicated as favorable polymeric loading materials for nanoparticle formulations, as they are biodegradable, non-immunogenic, and can be loaded with both hydrophilic and hydrophobic drugs, as indicated in Figure 4 [53].

Poly-(lactide-co-glycolide) used to formulate nanoparticles has previously not shown cytotoxicity in vitro or in vivo, and previous studies have also not found any significant adverse effects [54]. PLGA nanoparticle formulations are also considered to be eco-friendly and can be loaded with plant extracts and/or pure compounds. In a study carried out by Mahboob et al. (2020), a pure compound isolated from *Leea indica* (Burm. f.) Merr, namely gallic acid, was incorporated into a PLGA nanoparticle and tested against *Acanthamoeba triangularis* [55]. The nanoparticle exhibited a 7% increase in inhibition against trophozoites, and it also showed reduced cytotoxicity towards MRC-5 cells (IC_50_ = 30 µg/mL) when compared to gallic acid alone (IC_50_ = 10 µg/mL) [55]. This provides an opportunity for innovative natural product development that investigates currently under-explored delivery systems for TB research.

### 1.5. Limitations of PLGA Nanoparticles

The synthesis of PLGA nanoparticle formulations of 7-MJ could be a promising adjunct TB drug due to the previously reported activity of 7-MJ against *M. tuberculosis* and *M. smegmatis* and the potential of the PLGA nanoparticle formulation to decrease the cytotoxic effect of 7-MJ. Entrapping the hydrophobic pure compound 7-MJ into PLGA nanoparticles to administer as an adjunct TB drug could potentially aid the patient’s immune system to fight infection. This could serve as a basis for novel drug contenders that may assist in the development of drugs that will aid the current drug regimen [40,55].

However, there are many efficacy and safety evaluations that need to be carried out on PLGA nanoparticle formulations before clinical use approval can be obtained. Some current challenges that are faced when preparing PLGA nanoparticle formulations, which highlight the above statement, are poor drug release kinetics and drug entrapment efficiency [56]. These parameters are important for efficiently delivering drugs to the targeted cells and can be affected by an initial burst release of the encapsulated drug, which is the biggest challenge when developing PLGA nanoparticles [57]. This was exhibited in a study performed by Roberts et al. (2020), where within the first 3 days the PLGA nanoparticle exhibited an initial release of approximately 50% of the incorporated drug, Connexin43 mimetic peptide, with 73% of the drug released in vitro over 3 weeks [58]. In another study, the release profiles of a PLGA nanoparticle containing the hydrophobic drug curcumin were reported. In this study, the nanoparticle exhibited a 20–30% initial burst release in the first several hours before releasing up to 70% in a more linear fashion for about 18 days [59]. It is mostly preferable to not have any initial burst release of the drug within the first hours of the nanoparticle being administered, as this can cause some toxic effects, but rather a sustained linear release of the drug over an extended period of time. The initial burst release may make these nanoparticles unsuitable for certain therapeutic applications or drugs where a sustained and controlled drug release is required, such as the targeted delivery of drugs with half-lives of less than 3–4 h.

The variability in particle size can also impact the drug loading, release kinetics, and targeting efficiency of PLGA nanoparticles. The particle size as well as polymer composition and molecular weight also affect the biodegradation rate of nanoparticles, which makes it challenging to predict and control the release of encapsulated drugs [60]. The biodegradation of PLGA nanoparticles produces lactic acid and glycolic acid, which can lower the local pH and potentially impact surrounding tissue, which may potentially lead to tissue inflammation and other complications [61]. Immunogenicity is another concern, as PLGA nanoparticles may trigger an immune response in some cases, potentially resulting in inflammation or other adverse reactions. The extent of this response can vary based on factors like nanoparticle size, surface charge, and surface modification [62].

Furthermore, not all drugs are compatible with PLGA due to differences in solubility and chemical properties. This inherent limitation restricts the range of drugs that can be effectively encapsulated and delivered using PLGA nanoparticles [63]. Stability issues also come into play. PLGA nanoparticles are sensitive to various environmental factors, including temperature, humidity, and pH [64]. Thus, maintaining proper storage conditions is essential to preserving their stability and preventing aggregation.

From a regulatory perspective, obtaining approval for PLGA nanoparticle-based drug delivery systems can be a challenging process due to the need for comprehensive safety and efficacy assessments and the standardization of manufacturing processes [65]. Scaling up the production of PLGA nanoparticles from laboratory-scale to large-scale manufacturing can be costly and complex. Maintaining consistent quality, size, and drug-loading capacity at scale is a significant challenge [66]. Moreover, the disposal and environmental impact of PLGA nanoparticles, especially in medical waste, have raised concerns. Their biodegradability raises questions about their potential effects on ecosystems [67,68].

Despite these challenges, researchers are diligently working to address these limitations through modifications in nanoparticle design, surface functionalization, and formulation strategies. The ongoing research and innovation in this area continue to expand the potential of PLGA nanoparticles in addressing complex healthcare challenges.

### 1.6. Motivation for Using PLGA Nanoparticles for the Drug Delivery of 7-Methyljuglone

Poly-(lactide-co-glycolide) nanoparticles offer an array of mechanisms and strategies to mitigate the side effects of certain toxic compounds, such as 7-MJ. One of their most impactful attributes is their capability for targeted delivery. These nanoparticles can be tailored to precisely target an intended site of action within the body. This precision allows for the localized delivery of a toxic compound to a specific tissue or cell type, thereby minimizing exposure to non-targeted areas, reducing the potential for systemic toxicity, and avoiding collateral damage to healthy tissue. These properties are especially relevant in applications such as cancer therapy and other active and passive targeted treatments [69,70].

Prior studies evaluating active targeting have highlighted that incorporating a targeting ligand onto the surface of nanoparticles significantly amplifies both drug cellular uptake and cytotoxicity potency in comparison to nanoparticles without conjugation [69,70]. An example of this was exhibited in a study conducted by Babu et al. (2017), where the average (%) cell viability of lung cancer (A549) cells treated with arginine–glycine–aspartic acid (RGD) antibody-conjugated PLGA nanoparticles (54.7%) was significantly lower than the average (%) cell viability of A549 cells treated with nonconjugated PLGA nanoparticles (65.9%). Furthermore, the study exhibited that after the A549 cells were treated with RGD antibody-conjugated PLGA nanoparticles containing FluTax (fluorescent paclitaxel Oregon green 488), the cells exhibited higher uptake when compared to nonconjugated PLGA nanoparticles containing FluTax. The uptake by the A459 cells was determined based on the fluorescence intensity of FluTax after the cells were incubated for 24 h with the respective treatments. The reduction in cell viability and increased fluorescence intensity indicated by the RGD antibody-conjugated PLGA nanoparticles demonstrated the selective absorption of targeted delivery compared to non-targeted delivery at the same dose of actives for cancer treatment [70].

Passive targeted delivery of ligand-modified drugs and drug delivery systems to less accessible regions is facilitated by the enhanced permeability and retention (EPR) effect [71]. This mechanism involves the favored accumulation of nanoparticles in tumors, attributed to increased vascular permeability and reduced lymphatic clearance in contrast to normal tissues [71]. In the context of antimycobacterial therapy, tissues affected by bacterial infection often exhibit increased vascular permeability due to inflammation; therefore, it is postulated that the EPR effect operates in such infected tissues [71]. Drugs such as 7-MJ that are not soluble in water have brief plasma half-lives and are prone to encountering challenges in pharmacokinetics [72]. Nanoparticles with prolonged half-lives are, therefore, more likely to accumulate in these infected tissues [71]. In a study conducted by Kalluru et al. (2013), the efficacy of controlled release by PLGA nanoparticles loaded with rifampicin was compared to free rifampicin using passive targeting [73]. The results revealed that rifamicin-loaded nanoparticles were more effective in eradicating *Mycobacterium bovis* BCG within the targeted cells, macrophages [73].

When compounds are incorporated into PLGA nanoparticles, they are shielded from the surrounding environment. This protective encapsulation can prevent the immediate contact of the toxic compound with tissues or cells. The protective encapsulation can also shield the compound from degradation, metabolism, or enzymatic breakdown. This protection can lead to a more prolonged duration of action and potentially a lower toxic effect. In some instances, PLGA nanoparticles can effectively slow down the clearance of a toxic compound from the body, extending its presence at a lower concentration [74]. This prolonged exposure often leads to a milder toxic response compared to a bolus administration.

The polymer, PLGA, undergoes hydrolysis in the body, breaking down into two distinct monomers, lactic acid and glycolic acid, which are naturally occurring in the body and therefore pose minimal toxicity. Typically, PLGA is produced through a catalyst-mediated random ring-opening copolymerization of lactic acid and glycolic acid, forming a linkage through ester bonds between the two polymers. Both constituent polymers are metabolized by the Krebs cycle and converted to pyruvate [75]. Due to the distinct intrinsic properties of the two polymers, the resulting copolymer, PLGA, exhibits varied characteristics, and its degradation rate is contingent on the ratio of monomers utilized. For instance, an increased ratio of lactide to glycolide leads to slower drug release rates due to heightened hydrophobicity and a reduced degradation rate [75]. This is attributed to the hydrophobic nature of poly-lactic acid and the hydrophilic nature of poly-glycolic acid, which facilitate easy degradation within the body. Additionally, a lower molecular weight in the polymer renders it more susceptible to both degradation and drug release. This controlled release lessens the concentration of the compound, spreading its effects over time and thus minimizing its toxicity [74,75]. However, the initial burst release of the drug, as previously mentioned, needs to be taken into consideration to ensure that the nanoparticles are suitable for their intended therapeutic application.

Due to the hydrophobic nature and the likelihood of 7-MJ having a brief plasma half-life, as mentioned in the above discussion on targeted delivery, it also makes this drug and other hydrophobic drugs more likely to encounter challenges in pharmacokinetics [72,76]. These challenges include limited oral absorption through the gastrointestinal route and swift clearance from systemic circulation. Using PLGA nanoparticles as drug delivery systems for hydrophobic drugs can help them overcome the mucosal barrier and enhance their sustained release, therefore their bioavailability [76]. This was demonstrated in a study by Khalil et al. (2013), where the oral administration of PLGA nanoparticles containing the hydrophobic drug curcumin increased the drug’s mean half-life in rat plasma by approximately 4 h as well as a 2.9-fold increase in the drug’s bioavailability, as demonstrated by the maximum observed rat plasma concentration [77].

Poly-(lactide-co-glycolide) nanoparticles can also enhance the solubility of poorly water-soluble compounds, which also includes 7-MJ as it is a hydrophobic compound. This is a common challenge that PLGA nanoparticles address, as they have been shown to exhibit increased stability when suspended in biologic fluids [78]. This can help to ensure that the drug has more controlled and predictable absorption and distribution, which, in turn, reduces the likelihood of toxic overdosing. Furthermore, it is possible to customize some properties of PLGA nanoparticles, including size, surface charge, and surface modification, which in turn can optimize the delivery and release of the toxic compound. This tailoring process allows for the reduction of the compound’s overall toxicity [79,80]. In addition to these strategies, PLGA nanoparticles can be designed to co-deliver a toxic compound alongside a protective or synergistic agent. This dual-delivery approach not only counteracts the toxic effects of the compound but also enhances its therapeutic benefits [81].

However, it is crucial to acknowledge that while PLGA nanoparticles can effectively reduce the toxicity of certain compounds, the design and formulation of these nanoparticles must be meticulously considered based on the specific properties and pharmacokinetics of the toxic compound. The cytotoxicity of PLGA nanoparticles has been shown to be time- and/or dose-dependent [82]. In a study conducted by Di-Wen et al. (2015), PLGA nanoparticles incorporating LFC131 peptides were synthesized to improve the delivery of epirubicin to liver cancer tumors [83]. After 24 and 48 h of treatment, the nanoparticles demonstrated antiproliferative effects with IC_50_ values of 0.78 and 0.38 mg/mL, respectively, on human hepatic carcinoma (HepG2) cells [83]. Therefore, the observed cytotoxicity exhibited by these nanoparticles had a time-dependent pattern. Furthermore, in the development of folate-targeted PLGA nanoparticles for the specific delivery of doxorubicin to kidney fibroblast-like (COS-7) cells, dose-dependent effects were reported, as the cytotoxicity of these nanoparticles was directly proportional to the concentration of doxorubicin [82]. Rigorous preclinical and clinical studies are imperative to ensure the safety and efficacy of PLGA nanoparticle-based drug delivery systems when used to mitigate the toxic effects of compounds.

### 1.7. Clinical and Preclinical Applications of PLGA Nanoparticles

Poly-(lactide-co-glycolide) nanoparticles are broadly utilized in clinical experiments to treat or diagnose numerous diseases; however, there is currently no PLGA nanoparticle formulation treatment available on the global market. Attempts have been made for the transformation of PLGA-based nanoparticle formulations in a clinical sense to be used for medical treatment; one such example is Accurins. The Accurins project was intended to target an antigen expressed by prostate cancer cells and other types of solid tumors’ blood vessels, namely the prostate-specific membrane antigen (PSMA). The lead proprietary Accurin drug candidate, BIND-014, consisted of PEG-PLGA nanoparticles to transport docetaxel [84]. Phase II of the study on BIND-014 showed positive results; however, the development of nanomedicine was halted due to funds running low in late 2014 [85,86,87].

There are also other notable instances of successful clinical and preclinical applications of PLGA nanoparticles across a spectrum of medical domains, spanning cancer therapy, vaccines, targeted drug delivery, and treatments for diverse diseases. The ongoing expansion of successful clinical and preclinical applications underscores the ongoing innovation and ingenuity of researchers in addressing a broad array of healthcare challenges. An example of PLGA nanoparticle formulations for TB application was exhibited in a study conducted by Pandey et al. (2003), utilizing three first-line anti-TB drugs (isoniazid, rifampicin, and pyrazinamide). These nanoparticles were administered to experimental animals (guinea pigs) through nebulization [88]. In this study, the release kinetics and chemotherapeutic potential of the nanoparticles in animals infected with *M. tuberculosis* were evaluated. Rifampicin-loaded nanoparticles exhibited sustained drug release in plasma over 6 days, while isoniazid and pyrazinamide-loaded nanoparticles demonstrated sustained release over 8 days. In contrast, plasma levels of these drugs were detectable for only 12–24 h following oral or aerosol administration and for 6–10 h after intravenous administration [88]. The maximum concentration (Cmax) for nanoparticles loaded with rifampicin and pyrazinamide was comparable to that of orally administered standalone drugs. However, the Cmax for isoniazid-loaded nanoparticles exceeded that of their oral counterpart. The time to reach maximum concentration (Tmax) for drug-loaded nanoparticles was 24 h for rifampicin and 96 h for isoniazid and pyrazinamide [88]. The elimination half-lives for rifampicin, isoniazid, and pyrazinamide-loaded nanoparticles were determined to be 69.30 ± 4.00 h, 23.10 ± 2.00 h, and 69.00 ± 4.80 h, respectively. These findings indicated a substantial increase when compared to both oral and intravenous administration of the parent drugs. Enhanced absolute bioavailability values were observed for nanoparticles encapsulating rifampicin (6.50), isoniazid (19.10), and pyrazinamide (13.40), surpassing the bioavailability achieved through oral and intravenous dosing [88]. Remarkably, the nebulization of the drug-encapsulated nanoparticles for five doses resulted in the absence of tubercle bacilli in the lungs of guinea pigs infected with *M. tuberculosis*. In contrast, to achieve a comparable therapeutic effect from the parent drugs, 46 daily doses through oral medication are required [88].

There are more than 60 PLGA-based drug products currently on the market; this primarily includes an in situ gel (e.g., Eligard), a solid implant (e.g., Zoladex and Ozurdex), and PLGA microparticles (e.g., Decapeptyl, Lupron Depot, Nutropin Depot, and Sandostatin) [89,90,91]. Only 19 of these drugs have already been approved by both the European Medicine Agency (EMA) and the Food and Drug Administration (FDA) [92,93]. Only one PLGA nanoparticle drug formulation is currently enlisted on the clinicaltrials.gov website for bacterial infection; however, there was no TB treatment among the approved nanomedicines or upcoming trials. The enlisted nanoparticle formulation was tested for its activity against *Enterococcus faecalis* (*E. faecalis*). In 2020, PLGA nanoparticles coated with Chitosan polymer were prepared for this trial and incorporated into an in-situ gel for injection into the root canals of patients suffering from endodontic bacterial infection caused by *E. faecalis*. In 2022, the surface of the PLGA nanoparticles was modified and tested against resistant *E. faecalis* [94]. Although there are limited preclinical and clinical evaluations of PLGA nanoparticles, they remain valuable in the field of drug delivery and nanomedicine, as in the clinical and preclinical applications they have been involved in, they have shown great potential.

## 2. Discussion on the Challenges and Gaps in PLGA Nanoparticle Research

There has been a significant surge in interest in utilizing PLGA nanoparticles for drug delivery recently. Which is driven by their approved status by the FDA and the associated benefits [95]. Nonetheless, a major uncertainty persists regarding the safety of PLGA nanoparticles when carrying cytotoxic therapeutic agents [96]. Hence, it is crucial for researchers to develop PLGA nanoparticle formulations that not only possess biocompatibility, biodegradability, and cost-effectiveness but also exhibit the ability to release drugs sustainably, thereby minimizing systemic effects.

Determining the zeta potential and particle size of PLGA nanoparticles is essential. This is because an optimal nanoparticle size can facilitate the EPR effect for effective cellular drug uptake. It has been shown that with the increase in particle size, there is a corresponding increase in the IC_50_ of cytotoxicity [82]. This suggests that smaller nanoparticles result in lower IC_50_ values. This is, however, drug-dependent, as reported by Chiu et al. (2021) [82]. Despite hyaluronic acid leading to an increase in the particle size of nanoparticle formulations, it was demonstrated that this augmentation enhances the effectiveness of the encapsulated drugs in reaching the intended site [82]. An appropriate zeta potential, on the other hand, can enhance the stability of the nanoparticles. This enhancement can assist in ensuring the long-term stability and shelf-life of PLGA nanoparticles, which is also essential for their practical use [97]. Research on stabilizing formulations, preventing aggregation, and preserving drug integrity over extended periods is necessary.

Poly-(lactide-co-glycolide) nanoparticles have demonstrated effectiveness in pharmacological assays according to in vitro data in numerous studies. However, there is a notable absence of corresponding in vivo data for many of these studies. This lack of in vivo evidence raises concerns about the efficacy and safety of employing PLGA nanoparticles in human trials [98]. One such example is the difference in biological environments between in vitro and in vivo studies. In vitro studies often use cell culture systems with simplified environments, lacking the complexity of in vivo environments [99]. In vivo environments are accompanied by the presence of blood components, immune cells, enzymes, and other physiological factors. Therefore, in vivo environments can significantly impact the behavior and fate of PLGA nanoparticles [100]. To elaborate, in vitro studies might not comprehensively capture the systemic toxicity and side effects associated with PLGA nanoparticles. Systemic exposure and interactions with various organs can contribute to unexpected toxicities, which may not be evident in cell culture studies. In vivo studies provide exposure to more complex cell environments, addressing this limitation. [100]. A pharmaceutical formulation can only be deemed successful when both safety and efficacy are reliably ensured [101]. Therefore, there are many in vivo safety and efficacy evaluations that need to be carried out on PLGA nanoparticle formulations before clinical use approval can be obtained.

Furthermore, while PLGA nanoparticles have shown great promise in some preclinical studies, there is also a gap in translating this research into effective clinical applications [102]. More studies are required to bridge the gap between laboratory experiments and real-world clinical treatments. However, bridging this gap remains a significant challenge in various areas of study [103]. Some ways of potentially bridging the gap can be achieved by using relevant animal models that closely mimic human physiology and disease conditions. Additionally, incorporating diverse patient populations and employing advanced in vivo models will provide comprehensive insights into the safety and efficacy of PLGA nanoparticles [104,105,106]. The use of predictive models can also help in bridging this gap in translation. Using advanced computational models, such as in silico simulations and pharmacokinetic/pharmacodynamic modeling, can aid in predicting clinical outcomes based on preclinical data. Additionally, it can also assist in study design and decision-making before moving on to clinical applications [107]. Research focusing on regulatory approval could also play a role in bridging the translational gap. By initiating communication with regulatory agencies early in the development process, researchers can better understand the expectations of regulatory authorities. Researchers can also obtain guidance on study design and ensure that preclinical data align with regulatory requirements, streamlining the transition to clinical trials [108]. Researchers need to work closely with regulatory agencies to address the specific regulatory challenges associated with PLGA nanoparticle-based drug delivery systems. Compliance with safety and efficacy requirements is critical for clinical applications.

The long-term safety of PLGA nanoparticles, especially when used for chronic conditions or as carriers for sustained drug delivery, is another area that requires more investigation for the use of PLGA nanoparticles as delivery systems. It is essential to study potential accumulation, degradation products, and immune system responses over extended periods. Some current challenges that are faced when preparing PLGA nanoparticle formulations for drug delivery are poor drug release kinetics and drug entrapment efficiency. For PLGA-based nanoparticles, the mechanism and rate of drug release kinetics can be affected by the loaded drug by changing the breakdown between surface degradation and bulk degradation. The drug release of PLGA nanoparticles is, however, predominantly polymer-dependent rather than drug-dependent [109]. In surface erosion, the breakdown occurs at a constant pace at any time during the erosion. On the other hand, for bulk erosion, the breakdown is more complicated as there is no constant erosion pace. The breakdown does not occur for extended periods of time, after which it sets in spontaneously [110]. There are various factors that affect the drug-entrapment efficiency of PLGA nanoparticles. Some of them are due to the low solubility of the drug in the polymer matrix or the drug being incompatible with PLGA. This makes it difficult for them to be effectively entrapped. Another factor could be weak interactions between the drug and PLGA, which can lead to drug leakage during the nanoparticle formation process or subsequent storage [111]. The drug entrapment efficiency in PLGA nanoparticles can, however, be optimized by optimizing the formulation of the nanoparticles by considering the specific characteristics of both the drug and PLGA [112]. Characteristics of PLGA that can significantly influence encapsulation efficiency are the ratio of lactic acid to glycolic acid as well as the molecular weight of PLGA [113]. By selecting the appropriate ratio and molecular weight, the drug entrapment efficiency can be significantly improved. The choice of solvents for both PLGA and a hydrophobic drug, such as 7-MJ, can also affect the effectiveness of the drug entrapment [114]. Using a solvent that dissolves both the polymer and the drug facilitates uniform distribution and improves encapsulation efficiency. The use of co-solvent systems, for example, can enhance the drug’s solubility in the polymer solution, promoting better drug encapsulation. The use of co-solvents can also help to prevent drug precipitation during nanoparticle formation [115]. Incorporating surfactants or stabilizers (such as Tween-80, polyvinyl alcohol, sodium lauryl sulfate, and sodium dodecyl sulfate) during nanoparticle preparation can also aid in preventing drug aggregation as well as enhance the drug encapsulation efficiency [116]. Other potential ways of optimizing the drug entrapment efficiency of PLGA nanoparticles include systematically optimizing parameters. These parameters include the drug-to-polymer ratio, initial drug concentration, and polymer concentration to achieve the highest drug loading without compromising nanoparticle stability. The drug release kinetics and drug entrapment parameters of PLGA nanoparticles are important for efficiently delivering drugs to the targeted cells. Therefore, researchers should explore methods to maximize drug entrapment efficiency and control the release profile to achieve the desired therapeutic effect.

Some of the other gaps in the utilization of PLGA nanoparticles as delivery systems are in the evaluation of their impact on the environment. As PLGA nanoparticles may eventually reach the environment, it is important to investigate their potential ecological impact and develop strategies to minimize any adverse effects. Some environmental concerns include the degradation of PLGA, which results in the release of lactic acid and glycolic acid. In excess, these acids could alter the pH of the surrounding environment, potentially causing adverse effects [117]. A potential strategy to overcome this concern is the design of PLGA formulations that have controlled degradation rates. Additionally, evaluating and ensuring proper disposal methods for used PLGA-based materials can help mitigate the impact of these by-products on the environment. Other environmental concerns involving the production of PLGA nanoparticles include their potential to be overly energy-consuming and water-intensive. Significant energy consumption could contribute to greenhouse gas emissions, and intensive water usage could strain local water resources. To address energy consumption, it might be beneficial to implement energy-efficient manufacturing processes, such as utilizing renewable energy sources and optimizing production methods [118]. These implementations can help reduce the overall carbon footprint potentially associated with PLGA nanoparticle production. For the potential intensive water usage, it could be beneficial to employ water-saving technologies and adopt closed-loop water recycling systems in manufacturing facilities [118]. These systems and technologies can help minimize water usage and thereby reduce the nanoparticle’s environmental impact. By addressing the concerns through sustainable practices and responsible waste management, it is possible to minimize the potential adverse environmental effects associated with the use and production of PLGA materials. Ongoing research and development play a crucial role in achieving this goal.

Addressing these challenges and gaps in PLGA nanoparticle research will contribute to the development of more effective and safe drug delivery systems. This progress will ultimately benefit patients and advance the field of nanomedicine.

## 3. Conclusions

7-Methyljuglone was shown to be a promising candidate for incorporation into the current TB treatment regimen, given its notable antimycobacterial efficacy. However, its in vitro cytotoxic activity against various cancerous and non-cancerous cell lines raises concerns about its safety for TB patients. To ensure the safe and effective integration of 7-MJ into TB treatment, we propose the development of a PLGA nanoparticle delivery system that includes 7-MJ. Existing studies support the use of PLGA nanoparticle formulations to encapsulate 7-MJ, aiming to mitigate its cytotoxic effects. These studies indicate the absence of in vitro or in vivo cytotoxicity and the significant adverse effects associated with PLGA nanoparticle formulations. Nevertheless, several research gaps persist in understanding the broader application of PLGA nanoparticles as delivery systems, particularly concerning their clinical efficacy and safety. We recommend integrating more comprehensive preclinical and clinical assessments into research on the safety of PLGA nanoparticles as delivery systems. We also recommend that future studies incorporate thorough assessments of the interactions between PLGA nanoparticles and blood components, immune cells, enzymes, and other physiological factors. Addressing these aspects is crucial before confidently concluding that PLGA nanoparticles can reduce the toxic effects of compounds without contributing to toxicity or triggering adverse effects of their own.

## Figures and Tables

**Figure 1 pharmaceutics-16-00216-f001:**
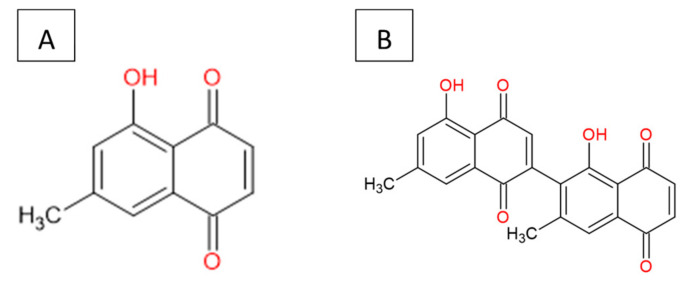
Chemical structure: (**A**) 7-methyljuglone and (**B**) diospyrin.

**Figure 2 pharmaceutics-16-00216-f002:**
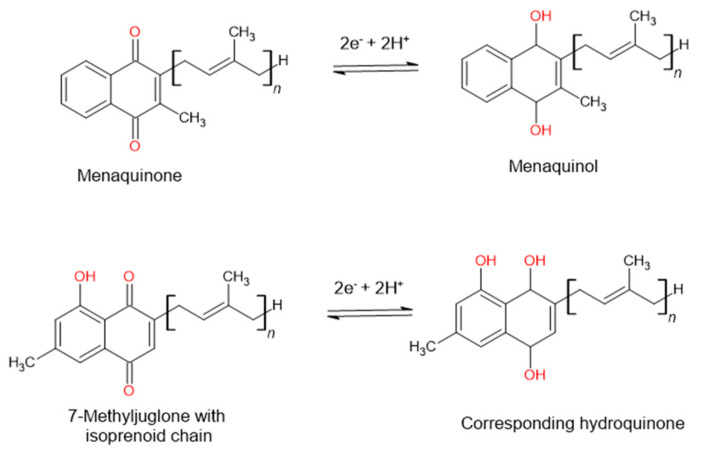
The postulated mechanism of action of 7-methyljuglone will disrupt the electron transport chain and therefore decrease or stop the electron flow in the bacterium, as suggested by Van der Kooy et al. (2006).

**Figure 3 pharmaceutics-16-00216-f003:**
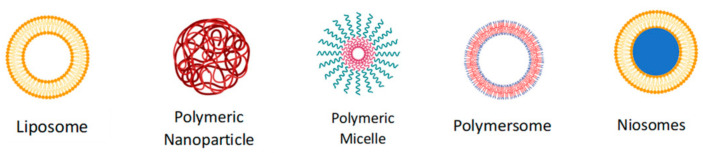
Examples of different nanocarriers explored for anti-TB therapy.

**Figure 4 pharmaceutics-16-00216-f004:**
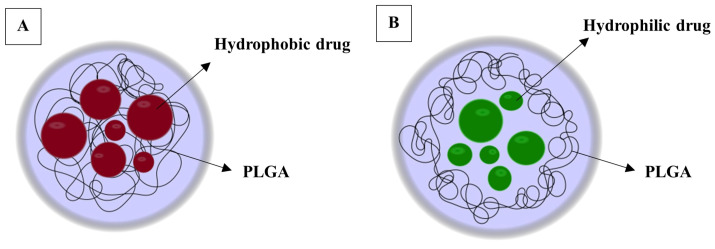
Examples of different drugs in nanocarriers: (**A**) hydrophobic drugs (Red) and (**B**) hydrophilic drugs (Green).

**Table 1 pharmaceutics-16-00216-t001:** First-line drugs currently used in conventional treatment [12,13,14].

Drug	Activity	Mechanism of Action	Side Effects
Ethambutol	Targets the cell wall of the mycobacterium through the biosynthesis of arabinogalactan.	Has shown great effect against growing bacterial bacilli.	Blurred vision, eyes have increased sensitivity to light, and pain behind the eyes.
Isoniazid	Only active against replicating metabolically active bacilli.	Isoniazid constrains the production of mycolic acids, a cell wall component that is very vital for *M. tuberculosis*, via the NADH ^1^-dependent enoyl-ACP ^2^ reductase, which the inhA gene encodes.	Hepatitis and skin rashes.
Pyrazinamide	Has effects against semi-dormant bacilli that can be found in acidic areas of TB lesions.	The drug converts pyrazinamide into pyrazinoic acid, which disrupts the membrane energetics of *M. tuberculosis*, which then causes the inhibition of membrane transport.	Arthralgia and hepatitis.
Rifampicin	Has activity against both non-growing and growing bacilli of mycobacteria.	Binding to RNA ^3^-polymerase via its β-subunit causes a hindrance in the elongation of the messenger RNA.	Nausea, vomiting, abdominal pains, hepatitis, and thrombocytopenic purpura
Streptomycin	Active against actively growing bacilli.	Inhibits the translation in protein synthesis of *M. tuberculosis.*	Damage to auditory and vestibular nerves, and sometimes renal harm.

^1^ NADH—nicotinamide adenine dinucleotide reduced form; ^2^ ACP—acyl carrier protein; ^3^ RNA—ribonucleic acid.

**Table 2 pharmaceutics-16-00216-t002:** The cytotoxic effect of 7-methyljuglone on cancerous and non-cancerous cell lines [27,35,37].

Cell Lines	IC_50_ (µM)
Cancerous Cell Lines	
Human breast cancer (MCF-7)	27.2
Immortal human (HeLa)	66.6
Spindle-shaped N-cadherin ^+^CD45^−^ osteoblastic (SNO)	81.4
Human prostate cancer (DU145)	11.9
Human oral epidermoid carcinoma (KB)	4.1
Human lung cancer (Lu1)	13.2
Hormone-dependent human prostate cancer (LNCaP)	3.7
Leukemia (HL60)	8.8
Non-cancerous Cell Lines	
Peripheral blood mononuclear cells (PBMC)	18.4
Human histiocytic lymphoma (U937)	Between 5.31 and 26.6
Umbilical vein endothelial cells (HUVEC)	5.7
Green monkey kidney (Vero)	80.4
Mouse macrophage (J774A.1)	20.8

## Data Availability

Data are contained within the article.

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
