# Peer review of "A Review of the Potential of Poly-(lactide-co-glycolide) Nanoparticles as a Delivery System for an Active Antimycobacterial Compound, 7-Methyljuglone"

_pharmaceutics, 2024, doi:10.3390/pharmaceutics16020216_

Round 1
Reviewer 1 Report
Comments and Suggestions for Authors
The review emphasizes the efficacy of PLGA nanoparticles for targeted drug delivery, citing studies that demonstrate enhanced cellular uptake and reduced toxicity in cancer treatment. It critiques the manuscript for insufficiently linking in vitro and in vivo data, urges a deeper exploration of long-term safety considerations, and highlights the need for practical solutions to bridge the gap between preclinical studies and clinical applications. Additionally, it calls for a more detailed examination of challenges specific to 7-Methyljuglone drug release kinetics in PLGA nanoparticles:
Comments for authors:
Title:
The title accurately reflects the content of the study and is concise and informative.
Introduction:
1. While the content is informative, consider organizing the section into subheadings to enhance the structure, such as Introduction, Impact of COVID-19 on TB, Current Challenges in TB Treatment, Introduction to Euclea natalensis A. DC., 7-Methyljuglone as a Potential Therapeutic Agent, and Mechanism of Action of 7-Methyljuglone.
2. Cytotoxic Effects and IC50 Values: The section on the cytotoxic effects of 7-MJ provides valuable information, including IC50 values for various cell lines. To enhance clarity, consider organizing the information into a concise table or figure for easy reference.
3. The section mentions that 7-MJ does not seem to have selectivity toward cancerous cell lines compared to non-cancerous cell lines. Consider discussing the implications of this lack of selectivity in the context of potential therapeutic applications.
4. The transition to the discussion of nanotechnology and its role in drug development is well-executed. To enhance the section, consider elaborating on specific examples or studies where nanotechnology has been successfully employed for anti-TB drug delivery.
5. Clarity in Targeted Delivery: The section discusses the precision of PLGA nanoparticles in targeted delivery, emphasizing their capability to minimize exposure to non-targeted areas. Consider providing a brief example or case study related to 7-MJ to illustrate the potential benefits of targeted delivery in the context of antimycobacterial therapy.
6. The section on PLGA polymer degradation provides valuable insights. To further enrich this discussion, consider exploring how the degradation characteristics of PLGA align with the pharmacokinetics of 7-MJ and contribute to its sustained release.
- Safety Concerns with Cytotoxic Agents: The manuscript appropriately highlights uncertainties regarding the safety of PLGA nanoparticles carrying cytotoxic agents, such as 7-MJ. However, it would be beneficial to delve deeper into specific safety concerns related to cytotoxicity, providing examples from relevant studies. This would offer readers a more nuanced understanding of the challenges associated with using PLGA nanoparticles for cytotoxic drug delivery.
- Linking In Vitro and In Vivo Data: While the manuscript rightly points out the lack of corresponding in vivo data for many studies, providing examples or insights into potential disparities between in vitro effectiveness and in vivo performance would strengthen this argument. Readers would benefit from a deeper understanding of the implications of this gap in the context of PLGA nanoparticle-based drug delivery.
- Bridging the Gap to Clinical Applications: The discussion on the gap between preclinical studies and clinical applications is crucial. To enhance this, the manuscript could offer suggestions or insights into potential strategies to bridge this gap effectively, addressing challenges in translation. This would provide readers with practical considerations for advancing PLGA nanoparticle research from the laboratory to clinical settings.
- Challenges in Drug Release Kinetics: While the challenges faced in drug release kinetics are well-stated, incorporating examples or case studies related to 7-MJ would illustrate how these challenges may impact the therapeutic efficacy of PLGA nanoparticles for this specific drug.
- Environmental Impact: The mention of evaluating the impact of PLGA nanoparticles on the environment is significant. Providing a discussion on specific environmental concerns and potential strategies for minimizing adverse effects would add depth to this aspect of the manuscript.
- Optimizing Drug Entrapment Efficiency: The discussion on factors affecting drug entrapment efficiency is informative. Adding specific recommendations or strategies for optimizing drug entrapment efficiency in PLGA nanoparticles, especially concerning 7-MJ, would be beneficial for researchers in the field.
Author Response
Please see the attachment for your consideration.

Reviewer 2 Report
Comments and Suggestions for Authors
I read this article with great interest, considering the topicality of the subject. The Introduction is too long and correctly written, with ample bibliography, and invites to go deeper into the manuscript. But the article does not contain any information about the preparation and operation of mixed PLGA/7-MJ particles, which was the supposed aim of a paper with the title of this one. The paper can only be considered when it includes data (preferably from the authors) on this 7-MJ delivery system, and not just a review on the potential uses of PLGA.
Comments on the Quality of English LanguageThe English is correct, and no errors have been detected.
Author Response

(The authors gave the same response as above.)

Reviewer 3 Report
Comments and Suggestions for Authors
Diedericks et al. studied about Poly-(lactide-co-glycolide) nanoparticles as a
delivery system for an active antimycobacterial compound, 7-methyljuglone. They collected lot of information. But, they need to address following comments before it get accepted.
Comment 1: Rewrite the abstract
Comment 2: Ln. 16, change Tuberculosis into tuberculosis
Comment 3: Ln. 17, remove M. tuberculosis in bracket, its understood.
Comment 4: Ln. 18, what is DC
Comment 5: In introduction, please give general induction about your topic.
Comment 6: In abstract, they mentioned about tuberculosis and Mycobacterium tuberculosis but in the Introduction section its missing.
Comment 7: Rewrite section 1.1, 1.2, 1.2.1 and 1.2.2
Comment 8: Its review, I dont know why Discussion section?
Comment 9: Rewrite the conclusion
Comment 10: Flow of manuscript missing in so many places, check grammar and typographical errors throughout the manuscript.
Comment 11: If possible, please give one simple and understandable schematic diagram for new readers.
Author Response

(The authors gave the same response as above.)

Round 2
Reviewer 1 Report
Comments and Suggestions for Authors
No further comments. Thank you!
Author Response
Please see attached the changes we incorporated according to the suggestions and recommendations of other reviewers.

Reviewer 3 Report
Comments and Suggestions for Authors
Comment 1: ln. 33, please remove (M. tuberculosis).
Comment 2: Please avoid two sentences as one paragraph in discussion section.
Comment 3: I dont know why discussion section in review article, instead give some heading.
Comment 4: Make conclusion as a single paragraph.
Author Response

(The authors gave the same response as above.)
